# Violence against Women: Attachment, Psychopathology, and Beliefs in Intimate Partner Violence

Iris Almeida [1,2,*], Carolina Nobre [2], Joana Marques [2] and Patrícia Oliveira [2]

1   Egas Moniz Center for Interdisciplinary Research (CiiEM), Egas Moniz Forensic and Psychological Sciences Laboratory (LCFPEM), Egas Moniz School of Health & Science, 2829-511 Almada, Portugal
2   Victims Information and Assistance Office (GIAV), Egas Moniz School of Health & Science, 2829-511 Almada, Portugal
*   Correspondence: ialmeida@egasmoniz.edu.pt

**Abstract:** Intimate partner violence (IPV) is a violation of women's human rights. Attachment is an IPV risk or vulnerability factor, in part because of the role that it plays in interpersonal relationships. An insecure attachment can predispose women victims to psychological maladjustment, which can reflect psychopathology. Likewise, our general society reveals beliefs that support and legitimize IPV. Thus, this study examined the relationship between adult attachment, psychopathology, and IPV beliefs. Our sample comprises 158 women IPV victims, aged between 18 and 73 years old. The psychological assessment tools Experiences in Close Relationships, Brief Symptom Inventory, and Scale of Beliefs about Marital Violence were used. All ethical issues had been taken due to the sensitive nature of the involved data. The results showed that most victims had a secure attachment style, and it may be an indicator that these women possibly had safe experiences with an attachment figure leading to reduced anxious state attachment in adult relationships. The results show a positive association between adult attachment and psychopathology, as well as between attachment and beliefs about violence. Globally, attachment is related to psychopathology and IPV beliefs. These results allow aid professionals and institutions to have deep knowledge about adult attachment as a risk or vulnerability factor for IPV and the adverse consequences of this phenomenon. IPV requires urgent attention since is the greatest threat to the health and safety of women around the world.

**Keywords:** violence against women; intimate partner violence; attachment; psychopathology; beliefs





## 1. Introduction

Violence against women, specifically intimate partner violence (IPV), a type of domestic violence (Alkan and Tekmanlı 2021), violates women's human rights worldwide (Soeiro et al. 2023; UN Women 2021; United Nations 1993; United Nations Secretary-General 2018; World Health Organization 2013, 2021a, 2021b). World Health Organization (World Health Organization 2021a) and United Nations Secretary-General (2018) estimates indicate that 26% of women, aged 15 and older, have been a victim of a current or former male intimate partner at least once in their lifetime. Since COVID-19, in a 2021 survey in 13 countries, 45% of women reported that they or a woman they know has experienced some form of violence (United Nations 2023). It is thus important to conduct studies to better understand women's victimization and develop strategies to reduce immediate and long-term consequences (physical and psychological), as well as explanatory theoretical perspectives of IPV.

The literature points out attachment theory as a model to explain adult (i.e., love) relationships because attachment is developed from experiences established throughout life. Attachment theory attempts to use a person's experiences of early relationships with caregivers to explain why some adults are more secure, resilient, or sensitive than others or the other way around. In this perspective, these psychological characteristics are indicative of how the individual's attachment system becomes organized throughout life, based on their experiences of attachment in childhood relationships. In this sense, some studies

describe a relationship between attachment and IPV (e.g., Hazan and Shaver 1987), because this theory can describe individual/personal differences (Barbaro et al. 2019) and helps to understand and predict future relational dynamics (Gormley 2005).

Research on attachment in adulthood has suggested that the quality of childhood is activated in intimate adult relationships during times of stress and plays an important role in this process. Thus, difficulties in attachment may be a parsimonious explanation for IPV (Mahalik et al. 2005; Ørke et al. 2021). Hazan and Shaver (1987) were pioneers in the use of attachment theory as a theoretical model that can explain the link between adult relationships and IPV since they argue that romantic love can be conceptualized as an attachment process. According to the model of these researchers, attachment can be evaluated as 1. Secure—well-being, the feeling of security, and trustworthy relationships; 2. Insecure/Avoidant—feelings of fear and discomfort concerning intimacy, as well as high self-reliance and the refusal of dependency on others; 3. Insecure/Anxious/Ambivalent— the constant need to create intimacy and intense concern with relationships and intimacy, fear of rejection and abandonment, jealousy or resentment, doubts about themselves, feeling less appreciated and understood by others.

Attachment is pointed out as an IPV risk or vulnerability factor (e.g., Doumas et al. 2008; Holtzworth-Munroe et al. 1997; Koral and Kovacs 2022; Sandberg et al. 2019; Smith and Stover 2016) because it focuses on the development models acquired during childhood and the role that plays in interpersonal relationships throughout the life cycle (Almeida et al. 2019; Roberts and Noller 1998), which can lead to difficulties in intimate relationships (Hazan and Shaver 1987; Holtzworth-Munroe et al. 1997; Rholes et al. 1998), and a higher probability of IPV (Dutton and White 2012; Dutton et al. 1994; Mahalik et al. 2005; Velotti et al. 2018), especially from the victim's perspective (Bonache et al. 2019), because women have a greater probability of being exposed to emotional and physical violence and have few protective factors.

Some researchers (e.g., Brennan and Shaver 1995; Feeney and Noller 1990; Silva et al. 2023) indicate that individuals with a secure attachment tend to experience more satisfaction and have trusting relationships, while individuals with an insecure attachment (i.e., expressed mainly as reluctance in the relationship and other mixed emotions, such as dependence, rejection, and feelings of fear) tend to experience high levels of anxiety, anger, and frustration in their intimate relationships. Thus, an insecure attachment can predispose women victims to psychological maladjustment by reducing their resilience and resources, which can reflect a greater vulnerability and psychopathology (Carnelley et al. 2016; Mikulincer and Shaver 2012). For example, victims that have a secure attachment show lower levels of psychopathology compared to victims with an insecure (e.g., avoidance) attachment (Pianta et al. 1996; Shurman and Rodriguez 2006), which present higher levels of psychopathology such as depression, anxiety, and anger (Scott and Cordova 2002; Shurman and Rodriguez 2006).

A secure attachment works as an internal resource, resulting in a better adaptation to stressful situations. IPV can directly affect the stability of women who have a secure attachment. On the other hand, it can also have an indirect effect on the stability of women who have an insecure attachment and tend to have moderate to high levels of anxiety attachment (Allison et al. 2008; Barbaro and Shackelford 2019; McClure and Parmenter 2020; Moreira et al. 2006) and likely affect how women perceive and interpret IPV (Weston 2008). International studies, such as Kuijpers et al. (2012), report that an avoidance attachment is a strong predictor of IPV victimization, for victims with average and high anger levels, and that women victims present higher levels of insecure attachment (Ponti and Tani 2019).

In general, studies demonstrated that there is a relationship between women's IPV victimization and insecure attachment (e.g., Doumas et al. 2008; Godbout et al. 2009; Grych and Kinsfogel 2010; Henderson et al. 2005; Ørke et al. 2021), because when women are exposed to IPV there is a greater tendency to feel anxious when creating social relationships as adults. Some studies (e.g., Bartholomew 1990; Bartholomew and Horowitz 1991; Brennan and Shaver 1995) also analyzed more specifically attachment styles: 1. Secure—comfortable

with intimacy and autonomy, tend to trust, less anxious, more resistant, fewer feelings of loneliness, greater social support, ability to solve problems in unstable periods, high self-esteem; 2. Insecure/Avoidant/Dismissing—difficulty in depending on the other, tend to have fewer stable relationships; 3. Insecure/Preoccupied/Anxious—concern about relationships (e.g., rejection, fear of not being loved), more feelings of loneliness, less social support, higher levels of stress and anxiety, tend to experience extreme emotions, low self-esteem, rely excessively on the acceptance of others; 4. Insecure/Avoidant/Fearful—tend to avoid closer relationships due to fear of rejection, governed by distrust and jealousy, greater vulnerability to loneliness, feelings of vulnerability and inadequacy, depend on the acceptance of others.

Based on attachment styles it was found that there is a prevalence of Insecure/Preoccupied, and in second place, Insecure/Fearful (e.g., Allison et al. 2008; Bookwala and Zdaniuk 1998; Henderson et al. 1997) among victims. For example, Henderson et al. (1997) conclude that more than 80% of victims had a negative internal model of self, more specifically a preoccupied style (53%) and a fearful style (35%), followed by a secure style (7%) and a dismissing style (5%). Victims with a fearful style may be more resistant to sharing their experiences, in contrast with those with a preoccupied style, who tend to leave their intimate partner more often.

Beyond attachment, psychopathology, and other factors that occur during the development of the human being and their interactions with significant people and with a set of events, potentiates the development of beliefs throughout life (Carlson and Worden 2005). It is therefore possible to verify that women victims, men offenders, and the general population reveal beliefs that support IPV and subsequently legitimize IPV (Carlson and Worden 2005; Copp et al. 2019; Neves and Almeida 2020). IPV beliefs are also influenced and are a consequence of a patriarchal culture, and traditions that emphasize male domination and female submission reinforce IPV (Neves and Almeida 2020). Although there exist a large number of studies on adult attachment (e.g., Godbout et al. 2009; Grych and Kinsfogel 2010; Henderson et al. 1997, 2005; Kuijpers et al. 2012) and psychopathology (e.g., Mikulincer and Shaver 2012; Pianta et al. 1996; Scott and Cordova 2002; Shurman and Rodriguez 2006), in Portugal there are no studies that analyze the relationship between IPV female victimization, adult attachment, psychopathology, and IPV beliefs. This study examined the relationship between adult attachment, psychopathology, and IPV beliefs in a sample of 158 IPV victims.

We hypothesize (Figure 1) that: 1. An insecure attachment can predispose women IPV victims to psychological maladjustment, which can reflect psychopathology (e.g., Carnelley et al. 2016; Mikulincer and Shaver 2012; Scott and Cordova 2002; Shurman and Rodriguez 2006). 2. An insecure attachment potentiates women IPV victims in the development of domestic violence beliefs (Carlson and Worden 2005; Copp et al. 2019; Neves and Almeida 2020).

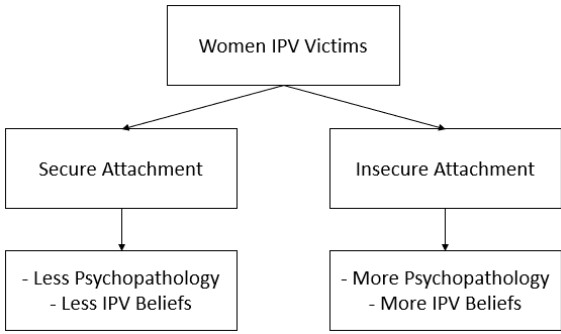

**Figure 1.** Relationship between adult attachment, psychopathology, and IPV beliefs.

## 2. Materials and Methods

### 2.1. Participants

We evaluate 158 female IPV victims aged between 18 and 73 years old (*M* = 43.95, *sd* = 12.01). Regarding nationality, most were Portuguese (86.7%), and the remaining nationalities were: Brazilian (5.7%), Angolan (1.9%), Cape Verdean (1.9%), Mozambican (1.3%), and others (2.4%—Romanian, Ecuadorian, English, and French). Concerning educational qualifications, they vary between 1st cycle and master's degree levels, and the professions of participants are also quite different (Table 1). The relationship between victims and offenders is: 54 married; 39 ex-boyfriends; 24 ex-spouses; 19 ex-partners; 16 partners; 4 boyfriends; 2 lovers.

**Table 1.** Sociodemographic characteristics.

|  | Frequency | Percentage (%) |
|---|---|---|
| Educational qualifications | | |
| 1st Cycle (1–4th year) | 28 | 17.7 |
| 2nd Cycle (5–6th year) | 24 | 15.2 |
| 3rd Cycle (7–9th grade) | 27 | 17.1 |
| Secondary Education (10–12th grade) | 25 | 15.8 |
| BSc Degree | 26 | 16.5 |
| Master | 14 | 8.9 |
| Professions | | |
| Leaders | 1 | 0.6 |
| Intellectual and scientific professions (e.g., lawyer, nurse, doctor, professor) | 25 | 15.8 |
| Technicians and professionals of intermediate level (e.g., nursing or doctor assistant, fitness instructor) | 13 | 8.2 |
| Administrative staff (e.g., secretary, receptionist) | 12 | 7.6 |
| Service and sellers (e.g., cook, hairdresser, police officer) | 38 | 24.1 |
| Agricultural, fishing and forestry workers (e.g., gardener) | 7 | 4.4 |
| Unemployed | 35 | 22.2 |
| Retired | 10 | 6.3 |
| Students | 13 | 8.3 |

### 2.2. Instrument

For this research, the following psychological assessment tools were used: Experiences in Close Relationships (ECR, Brennan et al. 1998; Portuguese version Moreira et al. 2006); Brief Symptom Inventory (BSI, Derogatis and Melisaratos 1983; Portuguese version Canavarro 1999, 2007); and Scale of Beliefs about Marital Violence (ECVC; Machado et al. 2007). We used the Portuguese Versions of all of the psychological assessment tools.

ECR is a 36-item self-report questionnaire based on the two-dimensional attachment system (anxiety vs. avoidance), and its purpose is to evaluate typical feelings and attachments in romantic relationships. Respondents used a 7-point, Likert-type scale ranging from 1 (disagree strongly) to 7 (agree strongly). Higher scores on the Anxiety (e.g., I worry about being abandoned) and Avoidant (e.g., I prefer not to show a partner how I feel deep down) subscales indicate higher levels of attachment anxiety (fear of rejection and abandonment, hypervigilance) and attachment avoidance (uncomfortable depending on the other), respectively. Higher concordance scores indicate lower levels of avoidance and concern in romantic relationships, ranging from a maximum of 252 to a minimum of 36. In the current study, Cronbach's alphas were good (0.80).BSI is a 53-item self-report measure in which respondents rate, between zero (not at all) and four (extremely), in the past week, various symptoms used to identify self-reported clinically relevant psychological symptoms, and covers nine symptom dimensions: somatization (e.g., faintness or dizziness), obsessive-compulsive (e.g., trouble remembering things), interpersonal sensitivity (e.g., your feelings being easily hurt), depression (e.g., thoughts of ending your life), anxiety (e.g., nervousness or shakiness inside), hostility (e.g., feeling easily annoyed or

irritated), phobic anxiety (e.g., feeling afraid in open spaces or on the street), paranoid ideation (e.g., feeling others are to blame for most of your troubles), and psychoticism (e.g., the idea that someone else can control your thoughts); and three global indices of distress: Global Severity Index, Positive Symptom Distress Index, and Positive Symptom Total. In the current study, Cronbach's alphas were excellent (0.97).

ECVC is a Portuguese self-report scale to assess beliefs about IPV and it is composed of 25 items, grouped into four factors: (1) legitimizing and trivializing of minor violence (e.g., insulting, slapping); (2) legitimization of violence by women's conduct (e.g., unfaithfulness, being a bad wife); (3) legitimization of violence by its attribution to external causes (e.g., alcohol consumption, financial difficulties); and (4) legitimization of violence by the preservation of family privacy (e.g., what goes on between a couple only concerns the couple). All items are rated on a Likert-type scale. Total scores can range from 25 to 125 points. The higher the scores obtained on the ECVC, the higher the levels of IPV legitimization. In the current study, Cronbach's alphas were excellent (0.94).

### 2.3. Procedure

Data were collected in the Victims Information and Assistance Office (GIAV) located at the Public Prosecutor's Office. The sample ($n$ = 158) came from IPV risk assessments, between 2014 and 2022, from semi-structured interviews, and from clinical and forensic assessment tools. IPV risk assessment can be defined as a process of collecting information about the people (e.g., offender's; victims; witness) involved to make decisions according to the risk of recurrence of violence (Almeida and Soeiro 2010).

All ethicaliissues had been taken due to the sensitive nature of the involved data and the respective informed consent, the confidentiality limits, and information about the ethics and technicians' impartiality were presented to participants. Informed consent was obtained from all subjects involved in the study. IPV victims signed an informed consent which contained the goal of the evaluation, the limits of confidentiality, and information about the ethics and impartiality of the technicians. Written informed consent has been obtained.

This study was conducted by rules defined by the Declaration of Helsinki, all ethical standards of scientific research were respected, as well as the Code of Ethics of the Order of Portuguese Psychologists and the General Data Protection Regulation. In addition to the above, the present study is included in the One Justice Project: The Forensic Psychology in Justice and Community approved by the appropriate institution.

### 2.4. Data Analysis

To analyze the data obtained, the IBM Statistical Version SPSS 0.28 program was used. Pearson correlations were performed between the scales and subscales used to verify the relationship between variables.

## 3. Results

This study examined the relationship between adult attachment, psychopathology, and IPV beliefs in a sample of women IPV victims.

The results show us that 112 victims (70.9%) had a secure attachment and 46 (29.1%) had an insecure attachment (assessed by two basic dimensions of individual differences in adult attachment style, namely avoidance and anxiety). Most victims had a secure attachment, and this may be an indicator that these women possibly had safe experiences with attachment figures, leading to reduced anxious state attachment.

We found a positive association between insecure attachment "Anxiety" and psychopathology (Table 2), namely Somatization, Obsessive Compulsive, Interpersonal Sensitivity, Depression, Hostility, Paranoid Ideation, Psychoticism, and Positive Symptom Total. We confirmed a relationship between adult attachment and psychopathology in this sample. For insecure attachment "Avoidance" we only identified a weak association with Phobic Anxiety and Positive Symptom Distress Index.

In this sense, we tried to understand better the relationship between attachment and psychopathology, and we found that Positive Symptom Total—TSP ($t = -2.139$; $p < 0.05$) is associated with an insecure attachment ($M = 27.08$, $sd = 16.17$), instead of a secure attachment ($M = 21.26$, $sd = 13.67$). We found an association between the Positive Symptom Distress Index—ISP ($t = 1.184$; $p < 0.05$) and secure attachment ($M = 225.13$, $sd = 66.67$), instead of an insecure attachment ($M = 93.01$, $sd = 64.38$). These results mean that victims may have a low ISP, indicating that the symptoms they have are not particularly intense and disturbing, especially if they have a secure attachment, while a victim who has a high TSP point to a complex constellation of symptoms, coupled with an insecure attachment.

**Table 2.** Correlation between attachment, psychopathology, and beliefs.

|  | Avoidance | Anxiety |
|---|---|---|
| Somatization (BSI) | 0.155 | 0.254 * |
| Obsessive Compulsive (BSI) | 0.136 | 0.265 * |
| Interpersonal Sensitivity (BSI) | 0.044 | 0.452 ** |
| Depression (BSI) | 0.230 | 0.429 ** |
| Anxiety (BSI) | 0.170 | 0.204 |
| Hostility (BSI) | 0.146 | 0.421 ** |
| Phobic Anxiety (BSI) | 0.257 * | 0.072 |
| Paranoid Ideation (BSI) | 0.133 | 0.432 ** |
| Psychoticism (BSI) | 0.174 | 0.328 ** |
| Global Severity Index (BSI–IGS) | 0.277 * | 0.131 |
| Positive Symptom Distress Index (BSI–ISP) | 0.281 * | 0.020 |
| Positive Symptom Total (BSI–TSP) | 0.198 | 0.377 ** |
| Factor 1 (ECVC) | 0.184 | 0.461 ** |
| Factor 2 (ECVC) | 0.236 * | 0.392 ** |
| Factor 3 (ECVC) | 0.245 * | 0.358 ** |
| Factor 4 (ECVC) | 0.284 * | 0.429 ** |
| Total Factor (ECVC) | 0.245 * | 0.442 ** |

Note: * $p < 0.05$; ** $p < 0.01$.

We also found a positive association between adult attachment and beliefs about violence (Table 2), namely minimizing minor violence, supporting violence through women's misconduct, supporting violence to an external cause, supporting violence through family privacy, and the general level of tolerance/acceptance of physical and psychological violence. We confirmed a relationship between adult attachment and IPV beliefs in this sample. However, when we tried to understand better the relationship between attachment and beliefs about violence, we did not find significant differences regarding the type of attachment (insecure vs. secure).

## 4. Discussion

This study examined the relationship between adult attachment, psychopathology, and IPV beliefs in a sample of IPV victims. The data demonstrated that the majority of IPV victims (70.9%) had a secure attachment, and this may be an indicator that these women possibly had safe experiences with attachment figures, preceding to reduced anxious state attachment. However, when women IPV victims develop an insecure attachment (29.1%) in their relationships, this manifests through an intense concern with those same relationships, a constant desire for closeness, and an obsession with abandonment and loss of intimacy. This study hypothesizes that an insecure attachment can predispose women IPV victims to psychological maladjustment, which can reflect psychopathology (e.g., Carnelley et al. 2016; Mikulincer and Shaver 2012; Scott and Cordova 2002; Shurman and Rodriguez 2006). In a global analysis, it appears that attachment is related to psychopathology, but also to IPV beliefs. An insecure attachment potentiates women IPV victims to the development of domestic violence beliefs (Carlson and Worden 2005; Copp et al. 2019; Neves and Almeida 2020).

These results allow aid professionals and institutions to have deep knowledge about adult attachment as a risk or vulnerability factor, as pointed out by Almeida et al. (2019) in a sample with IPV offenders and other studies (e.g., Doumas et al. 2008; Holtzworth-Munroe et al. 1997; Koral and Kovacs 2022; Sandberg et al. 2019; Smith and Stover 2016). Like the literature that analyzes the relationship between attachment and IPV, the results of the present study make perfect sense since women who have an insecure attachment are described as emotionally dependent, which is consistent with the notion that they have attachment problems (Dutton et al. 1994; Hazan and Shaver 1987; Henderson et al. 2005), requiring better individual understanding to predict future relational dynamics (Barbaro et al. 2019; Gormley 2005). As we see, an insecure attachment can predispose women victims to psychopathology (Carnelley et al. 2016; Mikulincer and Shaver 2012; Scott and Cordova 2002; Shurman and Rodriguez 2006) and, on the other hand, individuals with a secure attachment have lower levels of psychopathology (Pianta et al. 1996; Shurman and Rodriguez 2006). Beyond attachment and psychopathology, we found a relationship between attachment and the beliefs of individuals that support IPV, and subsequently legitimize IPV (Carlson and Worden 2005; Copp et al. 2019; Neves and Almeida 2020). IPV beliefs are also influenced and are a consequence of a patriarchal culture that emphasizes male domination and female submission, which reinforces IPV (Neves and Almeida 2020).

Despite the results obtained, we are aware that this investigation has some limitations, namely the wide age range of IPV victims (from 18 to 73 years old), which prevented us from analyzing in a more systematic and in-depth way the relationship between the variables studied. We also highlight that the sample is smaller than in other international studies, which may influence the results, which prevent us from drawing more valid data analysis and conclusions.

Another limitation is the evaluation of attachment by self-reported measures, which has been criticized since these do not cover all information processing strategies. These criticisms are based on the weak correlations found between the attachment scales that characterize avoidance and anxiety and the weak correlations found between self-report measures (Shaver and Mikulincer 2004). Given the evidence that a secure attachment is related to marital satisfaction, quality, and functioning, it is reasonable to hypothesize that an insecure attachment plays a primary role in IPV. The link between IPV and insecure attachment seems to make sense, as individuals who have this type of attachment are often described as emotionally dependent, consistent with the notion that women IPV victims have attachment problems. However, in the present study it was found that most IPV victims present a secure attachment to the offender. These results can be explained by the self-reported methodology used in this research.

Despite the limitations, however, we believe that we have contributed, in theoretical and practical terms, to the knowledge and development of this area of research, which can be, in the future, developed through qualitative studies and the improvement of specific tools for assessing attachment in the context of violence. Our findings highlight the relationship between the studied variables, reinforcing other studies.

There is some support in the literature for an association between attachment and IPV. An insecure attachment style can increase the IPV risk through the development of dysfunctional communication models. Adult attachment is related to the way individuals express their emotions and the level of intimacy in love relationships. The degree to which individuals feel uncomfortable with closeness and the degree to which individuals worry about being abandoned have important implications for how they interact with their intimate partners and can contribute to the understanding IPV. Women IPV victims with an insecure attachment tend to be more tolerant of the violence, so they are also more vulnerable to remaining in the abusive relationship, because they tend to excuse the offender.

## 5. Conclusions

The family is the core structure of society, recognized as a foundation of well-being, but it is also often a source of suffering, especially for women. IPV is a phenomenon that illustrates this last point well, and it has a high prevalence (World Health Organization 2013, 2021b). Prevalence data indicate that this type of violence requires urgent attention; it is the greatest threat to the health and safety of women around the world, its increasing visibility is associated with the redefinition of gender roles, the construction of a new social consciousness, and the affirmation of human rights. So why do we continue to excuse situations of violence? Perhaps, because we live in a patriarchal society, where the exercise of violence is legitimate, therefore it is imperative to assess the impact of, and to prevent and intervene more effectively in, these matters. In this sense, it is important to reflect on preventive and interventional policies, at the individual level, but also at the community and social levels, changing the paradigm through education for equality and non-violence, and breaking the public/private dichotomy. In especially troubled periods, such as those we are currently experiencing (e.g., pandemic, war, recession), which have occupied a prominent place in the media and our daily lives, it has never made so much sense to safeguard the Freedoms, Rights, and Guarantees of People, and especially of women, rebuilding a more sustainable society that is resilient and inclusive.

This research indicates the importance of implementing intervention programs with IPV victims to modify their patterns of relationships. Violence may reflect the behavioral models learned in the family. Direct exposure to parental violence can trigger insecure attachment models (Dutton et al. 1994), because it is expected that IPV victims who experienced or witnessed violence in childhood have more difficulties in developing a secure attachment in adult relationships. Because all research has implications, even if not immediate, for practice, it is important reflect on the implications of these studies, namely at the level of preventive and interventional policies necessary to reduce the occurrence of IPV, taking as an example an ecological model that includes: (a) education, therapeutic, and treatment programs; (b) identification and intervention plans in children who are affected by IPV, including articulation in the mental health system, more appropriate community responses, and articulation in schools; (c) interventional strategies at the clinical and educational levels with the objective of eradicating violence in the family context, including parental training and family therapy; (d) development of awareness campaigns and public education, highlighting the human and social costs of IPV, emphasizing the risk of death to women and including specific programs for children and young people to prevent violence; (e) systematic training for professionals in the justice system (e.g., police, prosecutors) and health and educational professionals); (f) continuous and permanent investigations, as more theoretical and empirical contributions are still needed.

**Author Contributions:** Conceptualization, I.A., C.N., J.M. and P.O.; methodology, I.A., C.N., J.M. and P.O.; formal analysis, I.A., C.N., J.M. and P.O.; investigation, I.A.; resources, I.A.; data curation, I.A., C.N., J.M. and P.O.; writing—original draft preparation, I.A.; writing—review and editing, I.A.; supervision, I.A.; project administration, I.A.; funding acquisition, I.A. All authors have read and agreed to the published version of the manuscript.

**Funding:** The authors thank FCT/MCTES—Foundation for Science and Technology, I.P., for the financial support to CiiEM (UIDB/04585/2020) through national funds.

**Institutional Review Board Statement:** This study was conducted in accordance with the Declaration of Helsinki, all ethical standards of scientific research were respected, as well as the Code of Ethics of the Order of Portuguese Psychologists and the General Data Protection Regulation.

**Informed Consent Statement:** Informed consent was obtained from all subjects involved in the study. IPV Victims signed an informed consent term, which contained the goal of the evaluation, the limits of the confidentiality, and information about the ethics and impartiality of the technicians. Written informed consent has been obtained.

**Data Availability Statement:** The data presented in this study are available on request from the corresponding author. The data are not publicly available due to the nature of the data.

**Conflicts of Interest:** The authors declare no conflict of interest.

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
