# Peer review of "Violence against Women: Attachment, Psychopathology, and Beliefs in Intimate Partner Violence"

_socsci, doi:10.3390/socsci12060346_

Round 1

Reviewer 1 Report

The article addresses an important issue. This study examined the relationship between adult attachment, psychopathology, and IPV beliefs in a sample of 158 female IPV victims aged 18 to 73 years. The authors provided a good theoretical framework and overview of the research and commented on the results obtained in relation to previous research and the broader social context.
Suggestions for improvement:

1/Introduction (line 50) "...but little is know..." - unclear; about what is little known?;

2/ Introduction (lines 73-74) „In general, studies demonstrated that is a relationship between women’s IPV victimization and insecure attachment…“ - sentence needs more clarity..."...that there is a relationship between..." or does the sentence refer to the previous one?

3/ Introduction (lines 75-77) „Some studies also analysed more specifically attachment styles and not just the dimensions (secure vs. insecure) and found that is a prevalence of an attachment style, namely preoccupied, and in second place, fearful…“ - unclear sentence

4/Procedure - The authors state that all ethical principles were followed in the research, but it would be important to indicate whether they obtained Ethical consent from the appropriate institution (responsible for the research) and whether participants signed an informed consent

5/ It is necessary to specify the data processing methods used

6/ Results (lines 154-156) “The results show us that 112 victims (70.9%) had a secure attachment and 46 (29.1%) had an insecure attachment (assessed by two basic dimensions of individual difference in adult attachment style, namely avoidance and anxiety).- the fact that 70.9% of the victims had a secure attachment contradicts the theories and research results presented by the authors in the introduction. It would be important to look back at these findings and try to explain this result (not to "ignore" this research finding).

7/ Discussion (lines 181-184) - “The data demonstrated that the majority of IPV victims (70.9%) had a secure attachment, and when women IPV victims tend to develop an insecure attachment (29.1%) in their relationships, which manifests through an intense concern with those same relationships, a constant desire for closeness, and an obsession with abandonment and loss of intimacy.” - sentence too long and thus unclear; it would be better to divide it into two sentences

8/Limitations of the study is also the wide age range of respondents, a relatively small sample to draw valid conclusions, and the very simple data processing methods used; for example, there is a great chance that the results are different for younger and older respondents, which would be expected given the changes in society (especially in terms of beliefs about IPV - awareness of human rights is greater today than 30/40 years ago) - consequently, this could affect other results (correlations with other constructs studied)

9/ Conclusion - it is written from the context of social responsibility and possible (necessary) strategies to prevent IPV, and that is excellent; however, what is missing is that the conclusion is more about the results of the study and what they mean for preventing IPV, working with victims, and possibly training professionals who work with IPV victims

10/Literature – it seems that the reference United Nations Secretary-General (2018) is not cited in the text  – please check

Minor editing of English language required

Author Response

Dear Reviewer,

We send you the revisions suggest about original research article entitled “Violence Against Women: Attachment, Psychopathology and Beliefs in Intimate Partner Violence”, for publication in Social Sciences – New Directions in Gender Research -2nd edition.

Thank you for your consideration of this manuscript and for your time and consideration!

We look forward to your response.

Sincerely,

The article addresses an important issue. This study examined the relationship between adult attachment, psychopathology, and IPV beliefs in a sample of 158 female IPV victims aged 18 to 73 years. The authors provided a good theoretical framework and overview of the research and commented on the results obtained in relation to previous research and the broader social context.
Suggestions for improvement:

Point 1: Introduction (line 50) "...but little is know..." - unclear; about what is little known?;

Response 1: “but little is know” is related with victim’s perspective. But in fact it is incomprehensible. In this sense, the authors chose to withdraw.

Point 2: Introduction (lines 73-74) „In general, studies demonstrated that is a relationship between women’s IPV victimization and insecure attachment…“ - sentence needs more clarity..."...that there is a relationship between..." or does the sentence refer to the previous one?

Response 2: The authors clarified the sentence as recommend by reviwer.

Point 3: Introduction (lines 75-77) „Some studies also analysed more specifically attachment styles and not just the dimensions (secure vs. insecure) and found that is a prevalence of an attachment style, namely preoccupied, and in second place, fearful…“ - unclear sentence

Response 3: We explain the specific attachment syles

Point 4: Procedure - The authors state that all ethical principles were followed in the research, but it would be important to indicate whether they obtained Ethical consent from the appropriate institution (responsible for the research) and whether participants signed an informed consent

Response 4: I explain better the procedure, namely that:

Data were collected in the Victims Information and Assistance Office (GIAV) located at Public Prosecutor's Office. The sample (n=158) is come from IPV risk assessments, between 2014 and 2022, from semi-structured interviews and clinical and forensic assessment tools. All ethical issues had been taken due to the sensitive nature of the involved data and the respective informed consent, the confidentiality limits, and information about the ethics and technician’s impartiality were presented to participants. Informed consent was obtained from all subjects involved in the study. IPV Victims signed an informed consent term, which contained the goal of the evaluation, the limits of confidentiality, and information about the ethics and impartiality of the technicians. Written informed consent has been obtained.

This study was conducted by the Declaration of Helsinki, all ethical standards of scientific research were respected, as well as the Code of Ethics of the Order of Portuguese Psychologists and the General Data Protection Regulation. In addition to the above, the present study is included in the One Justice Project: The Forensic Psychology in Justice and Community approved by the appropriate institution.

Point 5: It is necessary to specify the data processing methods used

Response 5: To analyze the data obtained, the IBM Statistical Version SPSS .28 program was used. Pearson correlations were performed between the scales and subscales used to verify the relationship between variables.

Point 6: Results (lines 154-156) “The results show us that 112 victims (70.9%) had a secure attachment and 46 (29.1%) had an insecure attachment (assessed by two basic dimensions of individual difference in adult attachment style, namely avoidance and anxiety).- the fact that 70.9% of the victims had a secure attachment contradicts the theories and research results presented by the authors in the introduction. It would be important to look back at these findings and try to explain this result (not to "ignore" this research finding).

Response 6: Most victims have a secure attachment, and it may be an indicator that these women had safe experiences with attachment figure lead to reduced anxious state attachment.

Point 7: Discussion (lines 181-184) - “The data demonstrated that the majority of IPV victims (70.9%) had a secure attachment, and when women IPV victims tend to develop an insecure attachment (29.1%) in their relationships, which manifests through an intense concern with those same relationships, a constant desire for closeness, and an obsession with abandonment and loss of intimacy.” - sentence too long and thus unclear; it would be better to divide it into two sentences

Response 7: I explain that the data demonstrated that the majority of IPV victims (70.9%) had a secure attachment, and it may be an indicator that these women possibly had safe experiences with attachment figure lead to reduced anxious state attachment. But when women IPV victims develop an insecure attachment (29.1%) in their relationships, they manifest through an intense concern with those same relationships, a constant desire for closeness, and an obsession with abandonment and loss of intimacy.

Point 8: Limitations of the study is also the wide age range of respondents, a relatively small sample to draw valid conclusions, and the very simple data processing methods used; for example, there is a great chance that the results are different for younger and older respondents, which would be expected given the changes in society (especially in terms of beliefs about IPV - awareness of human rights is greater today than 30/40 years ago) - consequently, this could affect other results (correlations with other constructs studied)

Response 8: Despite the results obtained, we are aware that this investigation has some limitations, namely the evaluation of attachment in IPV, the wide age range of IPV victims (from 18 to 73 years old), that which prevented us from analyzing in a more systematic and in-depth way the relationship between the variables studied. We also highlight that the sample is smaller than in other international studies, which may influence the results, which prevent us from drawing more valid data analysis and conclusions Despite the limitations, however, we believe that we have contributed, in theoretical and practical terms, to the knowledge and development of this area of research, which can be, in the future, developed through qualitative studies and the improvement of specific tools for assessing attachment in the context of violence. Our findings highlight the relationship between the studied variables, reinforcing other studies.

Point 9: Conclusion - it is written from the context of social responsibility and possible (necessary) strategies to prevent IPV, and that is excellent; however, what is missing is that the conclusion is more about the results of the study and what they mean for preventing IPV, working with victims, and possibly training professionals who work with IPV victims

Response 9: This research indicates the importance of implementing interventions programs with IPV victims to modify their patterns of relationships. Violence may reflect the behavioral models learned in the family. Direct exposure to parental violence can trigger insecure attachment models (Dutton et al., 1994), because it is expected that IPV victims who experienced or witnessed violence in childhood have more difficulties in developing a secure attachment in adult relationships. Because all research has implications, even if not immediate, for practice, it is important reflect on the implications of these studies, namely at the level of preventive and interventional policies necessary to reduce the occurrence of IPV, taking as an example an ecological model that includes: (a) education, therapeutic and treatment programs; (b) identification and intervention plans in children who are affected by IPV, including articulation with the mental health system, more appropriate community responses and articulation with schools; (c) interventional strategies at the clinical and educational level with the objective of eradicating violence in the family context, including parental training and family therapy; (d) development of awareness campaigns and public education, highlighting the human and social costs of IPV, emphasizing the risk of death of women and including specific programs for children and young people to prevent dating violence; (e) systematic training for professionals in the justice system (e.g. police, prosecutors) and health and educational professionals); (f) continuous and permanent investigations, as more theoretical and empirical contributions are still needed.

Point 10: Literature – it seems that the reference United Nations Secretary-General (2018) is not cited in the text  – please check

Response 10: I have in the references, but I forgot to out in the text, I already do that.

Reviewer 2 Report

This study examined the relationship between adult attachment, psychopathology, and IPV beliefs in a sample of women IPV victims. The researchers observed a majority of attachment among women who have experienced IPV, and they also showed an association between attachment, psychopathology, and IPV beliefs. It is an interesting topic and makes a relevant contribution to this area of knowledge. However, I have some comments that may be relevant to improving the manuscript.

Firstly, I suggest that the abstract be revised as the results are not entirely clear. Upon reading the introduction, I wonder if there is anything about disorganized attachment. Why has nothing been mentioned in this regard? If there is not enough evidence, I recommend that the authors indicate this.

Please indicate the study's design with a commonly used term in the title or abstract. At the end of the introduction, state specific objectives, including any prespecified hypotheses.

Were all instruments adapted and validated for the Portuguese population?

I am missing a section for data analyses that describe the analyses that were performed.

Considering the variability in the age of the sample, was age controlled?

Please explain how the study size was arrived at.

The sample (n=158) is said to come from IPV risk assessments. What does IPV risk assessment mean?

Please explain how missing data were addressed.

Regarding the discussion, I recommend that the authors reintroduce the objective and findings at the beginning.

The conclusion is barely as extensive as the discussion. The results need to be discussed in greater depth. Summarize key results with reference to study objectives.

Give a cautious overall interpretation of results considering objectives, limitations, multiplicity of analyses, results from similar studies, and other relevant evidence. Discuss the generalizability (external validity) of the study results.

Author Response

Dear Reviewer,

We send you the revisions suggest about original research article entitled “Violence Against Women: Attachment, Psychopathology and Beliefs in Intimate Partner Violence”, for publication in Social Sciences – New Directions in Gender Research -2nd edition.

Thank you for your consideration of this manuscript and for your time and consideration!

We look forward to your response.

Sincerely,

This study examined the relationship between adult attachment, psychopathology, and IPV beliefs in a sample of women IPV victims. The researchers observed a majority of attachment among women who have experienced IPV, and they also showed an association between attachment, psychopathology, and IPV beliefs. It is an interesting topic and makes a relevant contribution to this area of knowledge. However, I have some comments that may be relevant to improving the manuscript.

Point 1. Firstly, I suggest that the abstract be revised as the results are not entirely clear. Upon reading the introduction, I wonder if there is anything about disorganized attachment. Why has nothing been mentioned in this regard? If there is not enough evidence, I recommend that the authors indicate this.

Response 1: For this research we used: Experiences in Close Relationships (ECR, Brennan et al. 1998; Portuguese version Moreira et al. 2006) and according to this model, we have 1. Secure - comfortable with intimacy and autonomy, tend to trust, less anxious, more resistant, fewer feelings of loneliness, greater social support, ability to solve problems in unstable periods, high self-esteem; 2. Insecure/Avoidant/Dismissing - difficulty in depending on the other, tend to have fewer stable relationships; 3. Insecure/Preoccupied/Anxious - concern about relationships (e.g. rejection, fear of not being loved), more feelings of loneliness, less social support, more levels of stress and anxiety, tend to experience extreme emotions, low self-esteem, rely excessively on the acceptance of others; 4. Insecure/Avoidant/Fearful - tend to avoid closer relationships due to fear of rejection, governed by distrust and jealousy, greater vulnerability to loneliness, feelings of vulnerability and inadequacy, depend on the acceptance of others. And that’s why we don´t talk about disorganized.

I change abstract to results are clearer.

Point 2. At the end of the introduction, state specific objectives, including any prespecified hypotheses.

Response 2: This study examined the relationship between IPV, adult attachment, psychopathology, and IPV beliefs in a sample of 158 IPV victims. This study hypothesizes that: 1. An insecure attachment can predispose IPV victims to higher levels of psychopathology such as depression, anxiety, and anger (Scott and Cordova 2002; Shurman and Rodriguez 2006).

Point 3. Were all instruments adapted and validated for the Portuguese population?

Response 3. For this research, the following psychological assessment tools were used: Experiences in Close Relationships (ECR, Brennan et al. 1998; Portuguese version Moreira et al. 2006); Brief Symptom Inventory (BSI, Derogatis and Melisaratos 1983, Portuguese version Canavarro 1999, 2007); and Scale of Beliefs about Marital Violence (ECVC; Machado et al. 2007). We used the Portuguese Version for all the psychological assessment tools.

Point 4. I am missing a section for data analyses that describe the analyses that were performed.

Response 4. To analyze the data obtained, the IBM Statistical Version SPSS .28 program was used. Pearson correlations were performed between the scales and subscales used to verify the relationship between variables. 

Point 5. Considering the variability in the age of the sample, was age controlled? Please explain how the study size was arrived at.The sample (n=158) is said to come from IPV risk assessments. What does IPV risk assessment mean? Please explain how missing data were addressed.

Response 5. The age was not controlled, Data were collected in the Victims Information and Assistance Office (GIAV). The sample (n=158) is come from IPV risk assessments, between 2014 and 2022, from semi-structured interviews and clinical and forensic assessment tools. IPV risk assessment can be defined as a process of collecting information about the people (e.g., offender’s; victims; witness) involved to make decisions according to the risk of recurrence of violence (Almeida and Soeiro, 2010).

Point 6. Regarding the discussion, I recommend that the authors reintroduce the objective and findings at the beginning.

The conclusion is barely as extensive as the discussion. The results need to be discussed in greater depth. Summarize key results with reference to study objectives.

Response 6: I change discussion and conclusion according to orientations.

Reviewer 3 Report

I have carefully reviewed this manuscript and below is my decision.

- 158 responses were used in the analyses. This number may be sufficient to establish a structural equation model. However, you are making inferences about the population. Therefore, you need to use a sample to represent the population. Considering the population size, the sample size should be at least 384.

The article is incomplete in terms of basic statistical requirements. The current work is not suitable for publication because it has no adequate sample size.

I am happy having the opportunity to read and review the manuscript assigned to me, titled “Violence Against Women: Attachment, Psychopathology and Beliefs in Intimate Partner Violence” submitted to the Social Sciences.

I have carefully reviewed this manuscript and below is my decision.

- 158 responses were used in the analyses. This number may be sufficient to establish a structural equation model. However, you are making inferences about the population. Therefore, you need to use a sample to represent the population. Considering the population size, the sample size should be at least 384.

The article is incomplete in terms of basic statistical requirements. The current work is not suitable for publication because it has no adequate sample size.

Author Response

Dear Reviewer,

We send you the revisions suggest about original research article entitled “Violence Against Women: Attachment, Psychopathology and Beliefs in Intimate Partner Violence”, for publication in Social Sciences – New Directions in Gender Research -2nd edition.

Thank you for your consideration of this manuscript and for your time and consideration!

We look forward to your response.

Sincerely,

Point 1. 158 responses were used in the analyses. This number may be sufficient to establish a structural equation model. However, you are making inferences about the population. Therefore, you need to use a sample to represent the population. Considering the population size, the sample size should be at least 384. The article is incomplete in terms of basic statistical requirements. The current work is not suitable for publication because it has no adequate sample size.

Response 1. We have decided not to use statistical procedures as structural equation model, because this work intends to be exploratory taken due to the sensitive nature of the involved data. The IPV victims when goes to the prosecutor’s and Court signed an informed consent term, which contained the goal of the evaluation and the investigation and that’s why we have this sample size. Sometimes IPV victims don’t have psychological conditions when goes the court and comes to psychologists.

Reviewer 4 Report

Unfortunately, I found this paper hard to read - the use of references mid-sentence is unusual and somewhat disconcerting as one loses track of the themes - the paper reads in a very disjointed way as a consequence and loses any impact it could have.  I did not discern a clear discussion of the actual instrument used and the questions posed to the participants, so I think this aspect needs to be clarifed. 

Unfortunately, I found this paper hard to read - the use of references mid-sentence is unusual and somewhat disconcerting as one loses track of the themes - the paper reads in a very disjointed way as a consequence and loses any impact it could have.  I did not discern a clear discussion of the actual instrument used and the questions posed to the participants, so I think this aspect needs to be clarifed. 

Author Response

Dear Reviewer,

We send you the revisions suggest about original research article entitled “Violence Against Women: Attachment, Psychopathology and Beliefs in Intimate Partner Violence”, for publication in Social Sciences – New Directions in Gender Research -2nd edition.

Thank you for your consideration of this manuscript and for your time and consideration!

We look forward to your response.

Sincerely,

Point 1. Unfortunately, I found this paper hard to read - the use of references mid-sentence is unusual and somewhat disconcerting as one loses track of the themes - the paper reads in a very disjointed way as a consequence and loses any impact it could have. I did not discern a clear discussion of the actual instrument used and the questions posed to the participants, so I think this aspect needs to be clarifed. 

Response 1. We use references in mid-sentence because it’s the norms. We used following psychological assessment tools: Experiences in Close Relationships (ECR, Brennan et al. 1998; Portuguese version Moreira et al. 2006); Brief Symptom Inventory (BSI, Derogatis and Melisaratos 1983, Portuguese version Canavarro 1999, 2007); and Scale of Beliefs about Marital Violence (ECVC; Machado et al. 2007). We used the Portuguese Version for all the psychological assessment tools. And than we explain all the psychological assessment tools.

Round 2

Reviewer 2 Report

The authors have carried out most of the comments. However, the authors indicate that age has not been controlled. Given the variability in the age of the sample, I am interested to know why this variable was not take into account. What does the previous literature tell us regarding the relationship of age to the variables measured in this study?

Author Response

Dear Reviewer,

We send you the revisions suggest about original research article entitled “Violence Against Women: Attachment, Psychopathology and Beliefs in Intimate Partner Violence”, for publication in Social Sciences – New Directions in Gender Research -2nd edition.

Thank you for your consideration of this manuscript and for your time and consideration!

We look forward to your response.

Sincerely,

Response to Reviewer 2 Comments

Point 1. The authors have carried out most of the comments. However, the authors indicate that age has not been controlled. Given the variability in the age of the sample, I am interested to know why this variable was not take into account. What does the previous literature tell us regarding the relationship of age to the variables measured in this study?

Response 1. The variable age has not been controlled, but we analysed this variable (age), and we did not get significant results between age and attachment. However, we found significant results between age and beliefs. The literature said that older people have more legitimizing beliefs compared to younger people, but we chose not to put in the present research, because it was not the focus. We tried to examine the relationship between adult attachment, psychopathology, and IPV beliefs and not with age.

Reviewer 3 Report

I have carefully reviewed this manuscript and below is my decision.

-I would suggest adding to the literature and referencing it within the introduction and discussion as well.  There are studies that have examined intimate partner violence.

1) https://doi.org/10.1186/s12905-021-01333-1

It can be published after corrections are made.

Minor editing of English language required

Author Response

Dear Reviewer,

We send you the revisions suggest about original research article entitled “Violence Against Women: Attachment, Psychopathology and Beliefs in Intimate Partner Violence”, for publication in Social Sciences – New Directions in Gender Research -2nd edition.

Thank you for your consideration of this manuscript and for your time and consideration!

We look forward to your response.

Sincerely,

Response to Reviewer 3 Comments

Point 1. I have carefully reviewed this manuscript and below is my decision.

-I would suggest adding to the literature and referencing it within the introduction and discussion as well.  There are studies that have examined intimate partner violence.

1) https://doi.org/10.1186/s12905-021-01333-1

It can be published after corrections are made.

Response 1. We follow the suggesting of the reviewer and adding this study about intimate partner violence. But we do not interpret all the study, because the focus of our study is not sexual violence, but all type of intimate partner violence.

Reviewer 4 Report

Please have one more read through without the mark ups - I think there are a few plural references which should be singular 

Please have one more read through without the mark ups - I think there are a few plural references which should be singular 

Author Response

Dear Reviewer,

We send you the revisions suggest about original research article entitled “Violence Against Women: Attachment, Psychopathology and Beliefs in Intimate Partner Violence”, for publication in Social Sciences – New Directions in Gender Research -2nd edition.

Thank you for your consideration of this manuscript and for your time and consideration!

We look forward to your response.

Sincerely,

Response to Reviewer 4 Comments

Point 1. Please have one more read through without the mark ups - I think there are a few plural references which should be singular. 

Response 1. We use plural references there are several investigations that define the concepts in the same way or reach the same conclusions. For instance, when we talk about violence against women, we have a lot of institutions and authors that refers that violates women’s human rights worldwide, like UN Women 2021; United Nations 1993; United Nations Secretary‐General, 2018; WHO 2013, 2021, etc. When we talk about numbers and estimates it’s the same, like World Health Organization (WHO 2021); United Nations Secretary-General (2018); United Nations, 2023). For this reason, we have plural references.
